# Coastal-to-offshore submesoscale horizontal stirring enhances wintertime phytoplankton blooms in the ultra-oligotrophic Eastern Mediterranean Sea.

Yotam Fadida<sup>1,2</sup>, Vicky Verma<sup>3</sup>, Roy Barkan<sup>3,4</sup>, Eli Biton<sup>2</sup>, Aviv Solodoch<sup>5</sup>, and Yoav Lehahn<sup>1</sup>

<sup>1</sup>Department of Marine Geosciences, Charney School of Marine Science, University of Haifa, Israel

<sup>2</sup>Israel Oceanographic and Limnological Research, Israel

<sup>3</sup>Porter School of the Environment and Earth Sciences, Tel Aviv University, Israel

<sup>4</sup>Department of Atmospheric and Oceanic Sciences, UCLA

<sup>5</sup>Institute of Earth Sciences, the Hebrew University of Jerusalem, Israel

**Correspondence:** Yoav Lehahn (ylehahn@univ.haifa.ac.il)

**Abstract.** The large seasonal increases in marine photosynthetic organisms - i.e., phytoplankton blooms - are a ubiquitous oceanic phenomenon that contributes to the removal of carbon dioxide from the atmosphere and supports the growth of larger marine organisms. The underlying mechanisms controlling the intensity and timing of these blooms have been proposed to be dominated by vertical transport and mixing processes that are enhanced at fine-scale frontal and filamental circulations, com-

5 monly known as submesoscale currents. Here we show that the winter blooms characteristic of the ultra-oligotrophic waters of the Eastern Mediterranean Sea, which manifest as a seasonal increase in satellite-derived levels of surface chlorophyll, are also intensified by enhanced horizontal stirring induced by the submesoscale currents. Using ocean color remote sensing data and high-resolution numerical simulations, we demonstrate that the intensification of submesoscale currents in winter efficiently connect the coastal waters and the ultra-oligotrophic open-sea waters, thereby enriching the latter with chlorophyll-rich waters.

A climatological chlorophyll time series comparison between two different regions equidistant to the Nile River Delta indicates that this submesoscale horizontal stirring mechanism accounts for the ~24 % larger wintertime increase in surface chlorophyll observed downstream of the Nile Delta. These results shed new light on the processes governing phytoplankton bloom intensity and emphasize the important role of submesoscale horizontal stirring in modulating the marine ecosystem.

## 1 Introduction

Seasonal phytoplankton blooms occur worldwide, playing a critical role in the removal of carbon dioxide from the atmosphere (Arrigo et al., 2008; Kirchman et al., 1991), and in supporting the growth and development of organisms throughout the marine ecosystem (Frederiksen et al., 2006).

Over the past two decades, systematic study of phytoplankton blooms at the scale of ocean basins has become possible through satellite imaging of surface chlorophyll a (Chl), a proxy for phytoplankton biomass (Huot et al., 2007). Providing an 20 invaluable synoptic view of Chl distribution patterns, satellite Chl time series have revealed distinct geographical variability in bloom phenology, with contrasting responses to seasonal variations in vertical mixing (Yoder et al., 1993; Racault et al.,

2012; Behrenfeld and Boss, 2014; Silva et al., 2021). In the biologically productive mid- and high latitudes, Chl exhibits an abrupt increase during the springtime re-stratification of the water column (Trine Dale and Heimdal, 1999), whereas in the nutrient-depleted subtropics Chl shows a more moderate enhancement driven by intensified winter mixing (Siokou-Frangou et al., 2010; Barale et al., 2008).

Although these seasonal cycles are basin-scale in nature, they may be modulated by oceanic submesoscale currents—rapid, front- and filament-dominated flows with horizontal scales of 0.1–10 km and energetic vertical velocities (Mahadevan, 2016; McWilliams, 2016). Submesoscale motions can inject nutrients into the euphotic layer (Mahadevan, 2016; Lévy et al., 2018; Levy et al., 2023; Lathuiliere et al., 2011; Kessouri et al., 2020) and reorganize planktonic communities through lateral stirring and mixing (D’Ovidio et al., 2015; Lehahn et al., 2017; Lévy et al., 2018; Monte et al., 2013; Lehahn et al., 2018). Importantly, mixed-layer observations show that submesoscale flows themselves undergo a pronounced seasonal cycle, with markedly stronger activity in winter than in summer (Callies et al., 2015). Despite this emerging understanding, the relative contributions of vertical transport and horizontal stirring to shaping regional Chl distributions remain insufficiently quantified, particularly in oligotrophic environments.

The Eastern Mediterranean Sea (EMS) provides a natural setting to address this question. It is an ultra-oligotrophic basin with chronically low nutrient concentrations in the photic layer (Yacobi et al., 1995; Gitelson et al., 1995; Herut et al., 2000; Kress and Herut, 2001; Krom et al., 2014), where winter mixed-layer deepening is traditionally considered the dominant driver of phytoplankton seasonality (Hecht et al., 1998; Reich et al., 2022). Satellite Chl time series show a pronounced winter bloom (Barale et al., 2008; Salgado-Hernanz et al., 2019) and strong coastal-to-open sea gradients (Raveh et al., 2015), providing the spatial heterogeneity required for horizontal stirring to influence phytoplankton distributions. High-resolution simulations further indicate a strong seasonal modulation of submesoscale activity across the basin, with enhanced wintertime intensification (Solodoch et al., 2023; Verma et al., 2024), consistent with the observational evidence from other subtropical regions (Thompson et al., 2016).

Here, we study the role of submesoscale horizontal stirring in modulating the observed intensity of seasonal phytoplankton blooms in the EMS (Fig. 1A). Satellite datasets provide critical coverage of surface Chl and circulation in the region (Baaklini et al., 2024; Colella et al., 2016; Mayot et al., 2016; Amitai et al., 2010), yet their  $\sim$ 1-km gridded resolution remains coarser than the smallest dynamical scales relevant to submesoscale transport ( $\sim$ 0.1–10 km). Although newer high-resolution sensors such as Landsat and Sentinel-2 offer spatial resolutions of 10–60 m (Franz et al., 2015), their demonstrated applications for Chl retrieval have thus far been limited to coastal and estuarine environments (Yadav et al., 2019; Ogashawara et al., 2021; Poddar et al., 2019; Bonansea et al., 2017), with little validation in the open sea. Satellite altimetry likewise cannot resolve the submesoscale dynamics relevant here: the effective spatial resolution of conventional altimeter products in the Eastern Mediterranean exceeds  $\sim$ 50 km (Ballarotta et al., 2019), rendering them unsuitable for directly capturing the rapidly evolving kilometer- and sub-kilometer-scale motions that modulate surface chlorophyll. To bridge this gap, we combine multi-year satellite imagery of Chl concentration (1-km gridded) with a nested high-resolution numerical circulation model (3 km, 1 km, and 300 m horizontal resolution) that resolves the three-dimensional velocities and boundary layer turbulence needed to characterize the submesoscale motions shaping the satellite-observed Chl patterns.

## 2 Data and Methods

### 2.1 Data

**Satellite data.** This study is conducted using E.U. Copernicus Marine Service Information; DOI: 10.48670/moi-00300. The product used for estimating the mass concentration of Chl in seawater was the gridded, level 4 ocean-color daily 1-km data set for the years 2010-2020. This product combines data from several satellite missions (SeaWiFS, MODIS, MERIS, VIIRS-SNPP and JPSS1, OLCI-S3A) providing interpolated gap-free (Volpe et al., 2018) phytoplankton Chl concentration calculated using region-specific algorithms (Case-1 waters (Volpe et al., 2019), with new coefficients; and Case-2 waters (Berthon and Zibordi, 2004)).

Geostrophic currents were estimated from the Copernicus Level-4 gridded sea level anomaly (SLA) product, computed relative to a 20-year mean (1993–2012; DOI: 10.48670/moi-00142), at a spatial resolution of  $0.125^\circ$ .

**Biogeochemical model data.** To create the area-averaged nutriclines, we used the Copernicus global ocean biogeochemical hindcast product which uses the PISCES biogeochemical model. The product provides 3D biogeochemical fields at a 1/4 degree and on 75 vertical levels. DOI: 10.48670/moi-00019

**Haifa Section Cruise data.** Haifa Section Cruises have been conducted by the Israeli Oceanographic and Limnological Research Institute twice a year since 2002 (53 cruises to date). During the cruises physical, chemical, and biological data are collected along a transect of 8 stations, commencing in Haifa Bay and leading into the open sea perpendicular to the Israeli shelf. The cruise data has been used for the purpose of validating satellite and model data (see Fig. A1, A3). For in-depth cruise sampling and lab analysis methods see Ozer et al. (2017).

**Numerical Circulation Model** Coastal and Regional Ocean CCommunity (CROCO) model (Debreu et al., 2012), a version of the Regional Oceanic Modeling System (ROMS) (Shchepetkin and McWilliams, 2005) , has been used to simulate the EMS. A brief overview of the model is given here and we refer the reader to Solodoch et al. (2023) for a more comprehensive description.

The ocean model solves the free-surface, hydrostatic, Boussinesq primitive equations in a terrain-following vertical coordinate system. Furthermore, one-way grid nesting is used to generate three high-resolution, realistic regional simulations in the EMS with 3 km, 1 km, and 300 m horizontal grid resolutions, and with 80, 120, and 150  $\sigma$  levels, respectively [see Fig. 1 in Solodoch et al. (2023) for visualizing the nested-domain boundaries].

Atmospheric forcing is computed via the bulk formulae described in Fairall et al. (1996), and the atmospheric state is prescribed based on hourly output from the Integrated Forecasting System (IFS), which is a high horizontal resolution ( $\sim 9$  km) numerical weather prediction model. The turbulent mixing in the top and bottom boundary layers as well as the interior is parameterized using the K-profile parameterization (KPP; Large et al., 1994). The initial and open boundary conditions for the child solutions (1 km and 300 m) are obtained from the respective parent solutions (3 km and 1 km), following the methodology of Mason et al. (2010). For the 3-km simulation, the initial and boundary data are interpolated from the Copernicus Marine Environment Monitoring Service (CMEMS) Mediterranean Sea reanalysis 1/24 degree model output (Escudier et al., 2021).

The model has been extensively validated against observations, satellite data, and reanalysis data (Solodoch et al., 2023). The mean surface circulation pattern diagnosed with satellite altimetry data and the solution is in good agreement. Several known recurrent anticyclonic features appear consistently in the model solution. The surface geostrophic eddy kinetic energy variability derived from satellite altimetry is generally in agreement with that from the model. The mean SST from satellite observations and the mean SSS from CMEMS reanalysis data also compare well with those from the model solution. Furthermore, the 300-m model frequency spectra are in fair agreement with observations from the DeepLev mooring in the EMS and also capture the submesoscale energization in 1-5 day band during winter above 500 m depth. This research utilizes all the three nested solutions. The 3-km resolution model is run from January 2017 to December 2019, and the 1 km resolution model from February 2017 to December 2018. The 300-m resolution model ran for 53 days in the winter of 2018 (16-Jan to 09-Mar) and 69 days during the summer of 2018 (16-Jun to 23-Aug).

## 100 2.2 Methods

**Lagrangian particle tracking** For the Lagrangian particle tracking, the particles are modeled as material points that move with the local fluid velocity. The advection of the virtual particles is carried out over a 2D horizontal plane near the surface (2 m depth) using Parcels (Delandmeter and Sebille, 2019). The software uses a bilinear spatial interpolation of the surface model velocity to the particle location, and the time-stepping is performed with an explicit RK4 scheme. The particle trajectories are 105 computed during both winter and summer seasons, utilizing hourly velocity fields from the 300-m models. A total of 40000 tracer particles were released arranged in a  $200 \times 200$  grid pattern over a patch ( $31.5 - 32.0^\circ\text{N}$  and  $32.75 - 33.25^\circ\text{E}$ ) in the vicinity of the Nile Delta on January 23, 2018 and July 19, 2018 for the wintertime and the summertime analyses, respectively, and advected for 40 days during winter and 33 days during summer, until the end of the flow simulation. The particle advection during the summer is selected to coincide with a period of boundary current instabilities and spiral formation.

Along the particle trajectories, we also monitor flow properties such as relative vorticity,  $\zeta = (\partial v / \partial x - \partial u / \partial y)$ , and horizontal velocity divergence,  $\delta = (\partial u / \partial x + \partial v / \partial y)$ , where  $u$  and  $v$  are horizontal velocity components along zonal  $x$  and meridional  $y$  directions. The relative vorticity and horizontal divergence are first calculated with the model velocity and then linearly interpolated to the particle position.

**Mean offshore particle distance** To quantify the extent of offshore horizontal transport in the particle-tracking experiments, 115 we computed a mean offshore distance  $\bar{L}_y$  at each particle advection. Following the procedure illustrated in Fig. A4, we divided the domain between  $32.2^\circ$  and  $34.5^\circ\text{N}$  into zonally oriented strips of fixed width. For each strip, we identified the particle located farthest offshore and recorded its cross-shore distance from the EMS coastline. The mean offshore distance  $\bar{L}_y$  was then defined as the average of these farthest-offshore distances across all strips. The strip width was two grid cells (approximately 6 km) in the 3 km simulation and seven grid cells (approximately 2.1 km) in the 300 m simulations.

**Chlorophyll Climatologies** Monthly Chl climatologies were computed from the merged daily CMEMS 1-km product (2010–2019), and 95% confidence intervals were estimated as  $\pm 1.96 \times \text{SEM}$  (Standard Error of the Mean) using the Student's t-distribution. Chl concentration time series are calculated from daily data averaged spatially within the study region. The data is smoothed with a rolling average (30 day window).

**Probability density functions.** Horizontal Chl gradients were computed from the daily gridded satellite fields by first estimating the zonal and meridional spatial derivatives using finite-difference approximations on the native grid. The horizontal gradient magnitude was then calculated as

$$|\nabla \text{CHL}| = \sqrt{\left(\frac{\partial \text{CHL}}{\partial x}\right)^2 + \left(\frac{\partial \text{CHL}}{\partial y}\right)^2}. \quad (1)$$

Gradient values were spatially averaged within each analysis box (East and West) to produce daily domain-mean time series. Each seasonal subset (summer and winter) was converted into an empirical probability density function (PDF) by normalizing a 50-bin histogram to unit area. Statistical differences between regions were assessed using a two-sample Kolmogorov–Smirnov test applied separately to the summer and winter distributions.

To examine the dynamical forcing associated with submesoscale activity, we additionally computed PDFs of near-surface relative vorticity ( $\zeta/f$ ) from the 1 km numerical simulation, using the seasonally averaged values provided in the model output. Vertical velocity ( $w$ ) at 20 m depth and the parameterized vertical mixing coefficient  $A_{Kv}$  (from the K-Profile Parameterization scheme (Large et al., 1994)) were extracted along the same regional boxes, seasonally averaged, and converted to PDFs using the same normalization procedure.

**Chlorophyll enrichment factor** To quantify the offshore decay of Chl concentration downstream of the Nile Delta, we extracted a one-dimensional transect of Chl from the merged daily CMEMS 1-km product. The transect follows a south-east–northwest diagonal between (31.6°N, 34°E) and (33.5°N, 32.5°E), chosen because it provides the most monotonic and linearly increasing distance from the EMS coast; shifting the transect farther north or south would shorten the distance to another coastline and violate the desired single-coast geometry. Chl values were interpolated along this line and averaged by month, after which seasonal means were constructed for winter (January–March) and summer (July–September).

To compare seasonal enrichment independently of background concentrations, each transect was normalized by its far-offshore value. The enrichment factor was defined as:

$$E(x) = \frac{\text{CHL}(x)}{\text{CHL}_{\text{offshore}}}$$

For consistency across seasons,  $\text{CHL}_{\text{offshore}}$  was set to the minimum seasonal Chl concentration along the transect, which was 0.027 mg m<sup>-3</sup> in summer and 0.062 mg m<sup>-3</sup> in winter. Distances along the line were converted to kilometers using great-circle spacing. A 95% confidence interval was estimated at each position using the standard error and the Student's  $t$ -distribution, providing uncertainty bounds for both winter and summer enrichment profiles.

## 150 3 Results

### 3.1 Seasonality and spatial characteristics of satellite-derived Chl and modeled vorticity

Focusing on the vicinity of the Nile River Delta, at the southeastern part of the basin, which appears to be the largest Chl source in the area, we distinguish between two regions: the region to the east of the Delta, which is characterized by strong coast-to-open sea Chl gradient (Fig. A1), and the region to the west of the Delta, where such gradients are not observed (Fig. 1A,B).

This difference is attributed to the combined effect of nutrient enrichment from the Nile River Delta by agricultural seepage (Nixon, 2003), and the characteristic along-slope cyclonic circulation through the EMS boundary current, which transports the Chl-rich waters along the coast (Solodoch et al., 2023; Verma et al., 2024; Estournel et al., 2021; Gerin et al., 2009).

Comparing the 2010–2019 chlorophyll time series from two pelagic regions equidistant from the Nile Delta (East and West; Fig. 1C) reveals a pronounced seasonal asymmetry in surface Chl dynamics. During summer (July–September), Chl concentrations in the two regions are effectively identical, differing by only 0.1% (both  $\sim 0.028 \text{ mg m}^{-3}$ ). In contrast, during winter (January–March), the eastern region consistently exhibits higher biomass, with mean Chl concentrations 11.6% higher than in the western region (0.067 vs.  $0.060 \text{ mg m}^{-3}$ ). When referenced to the shared summer minimum, the wintertime increase in the East is  $\sim 24\%$  larger than in the West, indicating a stronger seasonal rebound east of the Nile Delta. Marked interannual variability is observed, with the winter East–West difference spanning approximately 3–27% across the decade ( $11.5 \pm 8.9\%$ )  
and the enhanced eastern seasonal increase ranging from  $\sim 5\text{--}65\%$  ( $25 \pm 22\%$ ). To evaluate whether the choice of domain size influenced the east–west comparison, we repeated the analysis using EN and WN boxes expanded by 50% in area while keeping their centers fixed. The resulting winter and summer Chl statistics, as well as the interannual variability, changed only marginally (e.g., winter East–West contrast: 11.6% vs. 10.9%; extra East winter rise: 24.95% vs. 22.66%). This demonstrates that the observed east–west differences are robust to reasonable variations in domain size, and are therefore not an artifact of  
the exact box boundaries.

In search of possible explanations for the winter-time difference between the two regions, daily images of satellite ocean color data are examined and compared to the modeled surface vorticity. The spatial characteristics of Chl display patterns that are qualitatively consistent with the modeled surface vorticity field (Fig. 2). Focusing on the area downstream of the Nile Delta, during summer (Fig. 2A,B), both fields display a distinct separation between the coastal region and the open sea, consistently generating a series of mesoscale (horizontal scales of about 10–100 km that last for days to weeks) meanders with relatively high Chl concentrations and high levels of vorticity that are constrained to the near-shore area. In contrast, during winter, the distinction between the coastal and open sea regions is substantially less pronounced and we observe small-scale variability in Chl concentrations and modeled vorticity throughout the entire EMS (Fig. 2C,D). However, while the Chl concentrations decay with increasing distance from the coast (Fig. 5, A1), the surface vorticity field is almost uniform throughout the basin. The  
similarities between the organization of the chl concentration and vorticity structures in the model solution further emphasize that the model captures the important seasonal circulation features in the region.

The seasonal change in the spatial characteristics of Chl concentrations and vorticity fields indicates a shift in the dynamical processes that shape the surface Chl distribution. The sharp contrast between the near-shore and open sea waters during summer indicates the presence of a transport barrier, which restricts the Chl-rich coastal waters from mixing horizontally with the open sea. This barrier likely results from the strong alongshore boundary current in the region (Rosentraub and Brenner, 2007). In this scenario, the dominant mode of offshore transport is via a chain of mesoscale eddies meandering along the coast, and the subsequent formation of Chl-rich submesoscale filaments at the eddy peripheries (Verma et al., 2024). In contrast, the more uniform distribution of surface Chl during winter suggests that the transport barrier weakens substantially. Here we test the

**Figure 1.** (A) Long-term summer (July–September) mean satellite-derived surface Chl ( $\text{mg m}^{-3}$ ) over the EMS, with geostrophic surface currents overlaid in white. Blue and orange boxes denote the eastern ( $32.5\text{--}33.5^\circ\text{N}$ ,  $33\text{--}34^\circ\text{E}$ ) and western ( $32.5\text{--}33.5^\circ\text{N}$ ,  $28.5\text{--}29.5^\circ\text{E}$ ) pelagic regions used for statistical comparisons. (B) Long-term winter (January–March) mean Chl and modeled surface currents for the same region. (C) Monthly climatology of Chl in the eastern (blue) and western (orange) regions; shaded areas denote 95% confidence intervals. Satellite data and modeled currents span 2010–2019.

hypothesis that this is due to an increase in submesoscale activity, which induces horizontal transport of Chl-rich water from the coastal region to the open sea.

Submesoscale currents are known to generate strong vertical motions that can enhance nutrient transport and, in some settings, stimulate phytoplankton growth (Mahadevan, 2016; Lévy et al., 2018). To evaluate whether such vertical processes could explain the downstream enhancement in wintertime open-sea Chl gradients, we compare the probability density functions (PDFs) of observed Chl-gradient magnitudes (Fig. 3A) with modeled indicators of submesoscale dynamics (Fig. 3B–D) in the western and eastern regions upstream and downstream of the Nile Delta. The Chl-gradient PDFs exhibit a clear seasonal and spatial asymmetry. During summer (warm colors in Fig. 3A), the East and West show similar distributions ( $\text{KS} = 0.242$ ,  $p = 0.009$ ), consistent with the uniformly oligotrophic and relatively stable surface conditions characteristic of the stratified season. In winter (cool colors in Fig. 3A), the distributions diverge sharply ( $\text{KS} = 0.678$ ,  $p = 6 \times 10^{-20}$ ), revealing that the

**Figure 2.** Representative spatial distribution patterns in snapshots of observed and modeled fields during summer and winter. (A) Summer surface Chl from satellite observations (30 July 2020). (B) Summer modeled surface relative vorticity (29 July 2018). (C) Winter surface Chl from satellite observations (25 February 2020). (D) Winter modeled surface relative vorticity (15 February 2018). The vertical black line in panels (B) and (D) marks the boundary between the 1 km and 300 m nested model grids.

eastern region experiences far more frequent and intense high-gradient events than the western region. To determine whether this winter asymmetry reflects differences in submesoscale dynamics, we examine modeled PDFs of relative vorticity, vertical velocity, and vertical mixing (Figs. 3B–D). Relative vorticity PDFs broaden markedly in winter, with standard deviation increasing by approximately fourfold and upper-tail values —corresponding to rare, high-magnitude vorticity events— rising by 300–525%, consistent with the known winter intensification of submesoscale currents (Barkan et al., 2019). This seasonal broadening occurs in both the eastern and western regions, indicating a basin-wide strengthening of submesoscale activity.

The modeled vertical velocity and vertical mixing fields provide an important check on whether vertical processes could explain the observed east–west asymmetry in Chl gradients. The winter vertical-velocity PDFs (Fig. 3C) show negatively skewed distributions in both regions, with  $|w| > 10^{-3} \text{ m s}^{-1}$ , a signature of intermittent submesoscale upwelling/downwelling (Mahadevan and Tandon, 2006), actually occurring more frequently west of the Nile Delta. Similarly, the KPP-derived vertical

mixing PDFs (Fig. 3D) indicate slightly stronger turbulent mixing in the western region during both seasons. Taken together, 210 these modeled fields do not exhibit the eastward enhancement that would be expected if vertical transport or mixing were responsible for the  $\approx 11\%$  higher winter Chl concentrations and steeper gradients observed downstream of the Delta. To verify that the east–west asymmetry does not arise from differences in the mixed-layer nutricline structure, we compared modeled 215 (PISCES) and observed nutrient profiles in both regions (Fig. A3), confirming that nutricline depth and nutrient availability, which are similar in both regions, cannot explain the observed Chl-gradient contrast. Taken together, these results demonstrate that neither enhanced vertical motions nor intensified mixing downstream of the Delta can account for the observed east–west contrast in wintertime Chl gradients. This suggests that vertical processes alone cannot account for the spatial structure of the 220 winter Chl field. Instead, as we show in the next section, the results are consistent with submesoscale horizontal stirring as the primary mechanism enhancing Chl-gradient magnitudes in the downstream (East) region. Accordingly, the westward decrease in Chl-gradient amplitudes can be understood as a dilution effect with increasing distance from the Chl-rich coastal waters discharged from the Nile Delta.

### 3.2 Lagrangian quantification of submesoscale horizontal stirring

The seasonal change in submesoscale horizontal transport and its impact on connectivity between the near-shore and open sea waters is emphasized by three Lagrangian particle-tracking experiments, one in summer and two in winter, initially seeded in the coastal region adjacent to the Nile Delta (red box in Fig. 4 A,D,G). During summer the particles are confined to the coastal 225 region, flowing through the EMS boundary current and the boundary current instability generated mesoscale eddies (Fig. 4 A-C). Conversely, winter is characterized by distinctly different trajectories, with particles being shed from the coastal region as early as five days after release (Fig. 4 D-F). While the general direction of flow is still alongshore, a substantial amount of particles make their way to the open sea through horizontal stirring by submesoscale motions that prevent the formation 230 of an effective transport barrier by the alongshore current. The 3-km simulation (Fig. 4 G-I) does not adequately resolve submesoscale features; therefore, it functions as a mesoscale-only control, enabling us to isolate the specific contribution of resolved submesoscale motions present in the 300-m winter run. Indeed, in the 3-km solution, the lateral transport is dictated 235 mainly by the anticyclonic boundary current eddies, very similar to what we observe in the 300-m summer simulation. However, the offshore particle distribution differs compared to the summer season because of the bigger size of the wintertime spirals, the further offshore position of the boundary current in winter (Verma et al., 2024), and the presence of partially resolved submesoscale filaments in the 3-km winter simulation. The difference in the particle distribution between the 3-km and 300-m 240 winter simulations is striking, with the strong impact of submesoscale stirring on the lateral transport clearly visible in the 300-m solution as the particles reach farther offshore than the extent of the anticyclonic spirals.

The extent of the offshore horizontal transport in the three model solutions is quantified by computing the mean offshore 245 distance ( $\bar{L}_y$ ) over an envelope encompassing the offshore particles between latitudes  $32.2^\circ - 34.5^\circ\text{N}$  (Fig. 5A, A4). The important contribution of submesoscale currents to the offshore horizontal transport during winter is emphasized by the fact 250 that  $\bar{L}_y$  increases continuously until more than 220 km in the 300-m winter solution, while constrained between 50-130 km and between 20-70 km in the 3-km winter solution and 300 km summer solution, respectively. The fluctuations in  $\bar{L}_y$  in the

**Figure 3.** Probability density functions (PDFs) of (A) satellite-derived Chl-gradient magnitudes (2010–2019), (B) modeled surface relative vorticity, (C) vertical velocity  $|w|$  at 20 m depth, and (D) KPP-derived boundary layer turbulence. Statistics are shown for the eastern and western pelagic regions north of the Nile Delta during summer and winter. Modeled fields are computed from the 1-km solution in both regions for consistency.

**Figure 4.** Lagrangian particle trajectories from the numerical simulations. (A–C) Summer simulation at 300 m horizontal resolution. (D–F) Winter simulation at 300 m horizontal resolution. (G–I) Winter simulation at 3 km horizontal resolution. Particles are initialized within the coastal region adjacent to the Nile Delta ( $31.5\text{--}32^\circ\text{N}$ ,  $32.75\text{--}33.25^\circ\text{E}$ ), shown by the red square in panels (A), (D), and (G). Snapshots are shown at days 5, 19, and 33.

latter two are due to the strong influence of the spirals. The larger  $\bar{L}_y$  in the winter 3-km solution compared with the summer 300-m solution (green and orange dots in Fig. 5A) may also be attributed to the generally further offshore location of the 245 boundary current in winter (Verma et al., 2024) (see also Fig. A2). To further estimate the regional impact of variations in submesoscale horizontal stirring, we examined the Chl enrichment factor (CEF) along the transect shown in Fig. 1A. The CEF transect (Fig. 5B) displays a pronounced seasonal difference. During summer, CEF values drop steeply within the first  $\sim 60$  km from shore, after which they approach offshore background levels. In winter, CEF remains elevated over a much broader region and shows little decline until nearly  $\sim 200$  km offshore. This seasonal contrast highlights a substantially wider downstream 250 footprint of coastal Chl during winter compared to summer. The  $\sim 60$  km sharp decrease in CEF during summer is consistent with the boundary current width during this season (Verma et al., 2024).

#### 4 Discussion and Conclusions

Combining high-resolution numerical modeling and ocean-color satellite observations, we elucidate the role of submesoscale horizontal stirring in shaping the phenology and spatial structure of phytoplankton blooms in the oligotrophic waters of the 255 EMS. The statistical analysis of a decade of satellite-derived Chl fields reveals a robust seasonal asymmetry east and west of the Nile Delta, with wintertime Chl concentrations and gradient magnitudes consistently elevated downstream of the coastal nutrient source. Notably, the magnitude of this enrichment exhibits substantial interannual variability, with wintertime East–West differences ranging from only a few percent to more than 25%, indicating that the strength of submesoscale-driven connectivity varies considerably from year to year. By contrasting these patterns with modeled vorticity, vertical velocities, and mixing 260 fields, we show that the observed winter enhancement cannot be explained by local vertical processes or regional differences in nutricline structure. Instead, the results point to strengthened submesoscale horizontal stirring during winter as the primary mechanism that increases coastal–open sea connectivity and supplies Chl-poor pelagic waters with Chl-rich coastal waters.

Our findings indicate that this seasonal intensification in horizontal stirring drives a basin-scale regime shift—from a relatively isolated coastal region during summer, bounded by a strong transport barrier, to a well-connected system during winter 265 in which submesoscale filaments and fronts efficiently redistribute materials offshore. Because this enhancement coincides with the basin-wide winter phytoplankton bloom, the increased stirring amplifies the bloom’s magnitude through purely physical pathways, independent of biological production. Submesoscale currents can affect phytoplankton blooms through several physical mechanisms, including enhanced vertical transport and changes in turbulent mixing that influence phytoplankton residence within the euphotic zone (Mahadevan, 2016; Lévy et al., 2018). Here, we highlight an additional pathway in which 270 submesoscale horizontal stirring contributes to bloom enhancement by laterally redistributing chlorophyll-rich waters across strong coastal–offshore gradients. Such a mechanism is expected to be most relevant in oligotrophic regions adjacent to nutrient sources. As shown in previous works, phytoplankton blooms and the processes underlying them in the EMS are representative of those in oligotrophic subtropical gyres (Varkitzi et al., 2020; Biagio et al., 2022). Moreover, the Chl seasonality characterizing these regions, with elevated Chl levels associated with a relatively deep mixed layer during winter, is intrinsically linked to 275 the seasonality of submesoscale activity worldwide (Callies et al., 2015; Mensa et al., 2013; Brannigan et al., 2015; Thompson

**Figure 5.** (A) Mean offshore distance  $\bar{L}_y$  of particles in the summer 300 m, winter 300 m, and winter 3 km simulations. Offshore distance is computed over latitudes 32.2–34.5°N using the envelope definition described in Fig. A4. (B) Chl enrichment factor (CEF) along the transect shown in Fig. 1A,B for winter and summer. Shaded regions denote 95% confidence intervals. CEF is computed from 10 years of satellite-derived Chl (2010–2019).

et al., 2016; Zhan et al., 2022). In winter, when the mixed layer is deep and stores a great deal of available potential energy, mixed layer instabilities (Boccaletti et al., 2007; Fox-Kemper et al., 2008; Taylor and Thompson, 2023) and frontogenesis events (Capet et al., 2008; McWilliams et al., 2015; Barkan et al., 2019; Verma et al., 2019) are frequent, leading to the formation of an energetic submesoscale current field that drives intense lateral stirring. In contrast, the shallow mixed layer during 280 summer suppresses these processes, reducing the prevalence and strength of submesoscale currents. Therefore, submesoscale horizontal stirring is likely to enhance the intensity of winter phytoplankton blooms in locations where nutrient-poor waters exist in close proximity to nutrient-rich waters, such as the periphery of subtropical gyres.

Looking forward, the emerging SWOT mission will provide unprecedented spatial resolution of sea-surface height and may help identify submesoscale features and validate high-resolution model output. However, its low temporal sampling—with revisits on the order of two weeks—limits its ability to capture the rapid evolution or seasonal progression of submesoscale dynamics. SWOT is therefore expected to serve as a complementary, event-focused tool rather than a basis for continuous phenological analysis.

These findings underscore the importance of submesoscale horizontal stirring as a climate-sensitive driver of bloom dynamics in oligotrophic regions and highlight the need for integrated observing–modeling approaches to unravel fine-scale physical–biological coupling in the ocean.

## Appendix A: Supplementary Figures

**Figure A1.** Comparison of Chl ( $\text{mg m}^{-3}$ ) concentrations measured during the Haifa Section Cruises (averaged 2002-2020) and satellite data from Copernicus (averaged 2010-2019) at 9 stations with increasing distances from the coast. Both in-situ and satellite measurements show the same general trend with higher concentrations by the coast that gradually drop with distance from the coast. The decrease rate during the winter slows down at approximately 55 km from the coast and levels out while during the summer the CHL values continue to decline. The error bars demonstrate how there is substantially more variability in the coastal measurements compared to the open sea.

**Figure A2.** PDFs of vorticity and horizontal velocity divergence associated with the offshore particles, defined as those that are more than 100 km away from the coast. The offshore particles are sampled from day 19 - day 40. The vorticity (horizontal divergence) PDF from 300-m simulation clearly shows positive (negative) skewness, suggesting that the offshore particles are associated with the submesoscale convergent cyclonic filaments. These features are also observed from the 3-km solution, as the vorticity PDF is positively skewed (skewness = 0.55) and the horizontal divergence negatively (skewness = -0.22). Nevertheless, the skewness magnitudes in the 3-km simulation is significantly smaller than the 300-m solution.

**Figure A3.** A-C comparisons of nutriclines ( $\text{NO}_3$ ,  $\text{PO}_4$ ,  $\text{Si}$ ) between the PISCES biogeochemical model averaged over the two study regions (2002-2019) and Haifa Section cruise data collected between 2002-2019 within the eastern region. The modeled nutrient profiles reproduced the observed nutricline structure with high fidelity ( $\text{NO}_3: r = 0.96$ ;  $\text{PO}_4: r = 0.79$ ;  $\text{Si}: r = 0.97$ ), indicating that the vertical shape and depth of nutrient gradients are realistically represented. However, Kolmogorov-Smirnov tests ( $p \ll 0.001$ ) reveal systematic offsets in concentrations, with the model tending to overestimate  $\text{NO}_3$  and  $\text{PO}_4$  and slightly underestimate  $\text{Si}$ . Comparison between the eastern and western model domains shows that the profiles are nearly identical in shape ( $r \geq 0.998$ ), with only small magnitude differences for  $\text{NO}_3$  and  $\text{PO}_4$ , and a somewhat larger shift for  $\text{Si}$ . Overall, these results demonstrate that the modeled vertical nutrient structure is robust across the basin and broadly consistent with observations, with remaining discrepancies arising primarily from concentration biases rather than differences in vertical structure.

**Figure A4.** Illustration of the computation of the mean offshore distance  $\bar{L}_y$  using the 3 km particle experiment. Shading indicates the zonally oriented strips between  $32.2^\circ$  and  $34.5^\circ$ N from which the farthest-offshore particle in each strip is identified. The definition of  $\bar{L}_y$  and the strip widths used in each model configuration are provided in the Methods section.

*Author contributions.* Y.F. compiled, processed, and analyzed the satellite data. R.B., A.S. and V.V. developed the numerical model. V.V. analyzed the model output. Y.L. and R.B. oversaw the research. Y.F., Y.L., V.V., A.V., R.B. and E.B. interpreted the results. Y.F. led the writing of the paper with contribution from all coauthors

*Competing interests.* The authors declare they have no competing interests.

*Acknowledgements.* We would like to acknowledge support from the Helmholtz International Laboratory: The Eastern Mediterranean Sea Centre- An Early-Warning Model-System for our Future Oceans: EMS Future Ocean REsearch (EMS-FORE), and from the Israel Science Foundation (grants 1266/23 and 2054/23). RB and VV would like to acknowledge support from the Israeli Ministry of Energy and from the Israel-US Binational Industrial Research and Development (BIRD) foundation. We would like to acknowledge the IOLR Chemistry  
Department for kindly providing us with the Haifa Section Cruise data.

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
