# Peer review of "Coastal-to-offshore submesoscale horizontal stirring enhances wintertime phytoplankton blooms in the ultra-oligotrophic Eastern Mediterranean Sea."

_EGUsphere, 2025_

## Referee Comment (RC1)

**Specific comments**

1. Title: Given the context of the manuscript, I understand the authors' intended meaning for the phrase "seasonal enrichment of low-chlorophyll surface waters". But, I think it could be clearer. The word "enrichment" describing the "low-chlorophyll surface waters" could be interpreted to mean that horizontal stirring makes chlorophyll even lower in the region. "Seasonal" is also vague, considering that the study focuses on winter blooms. Some alternative suggestions are:
   - "Coastal to offshore submesoscale horizontal stirring enriches wintertime phytoplankton blooms in the oligotrophic Mediterranean Sea"
   - "Submesoscale horizontal stirring increases offshore chlorophyll concentrations during the winter bloom in the Eastern Mediterranean Sea"

2. The authors tackle the challenge of explaining satellite chlorophyll observations with the dynamics observed in a numerical model. To increase trust that they're not comparing "apples and oranges", I suggest that they include more information about the validation of the numerical model and how representative it may be of real-world physics observed in the system. A few sentences would be sufficient, potentially citing more of the findings of the original study by Solodoch et al. 2023 (by the way, I think this study is accidentally cited as two separate citations in this manuscript). Additionally, an explanation for why a numerical model, as opposed to observations, was needed to capture the submesoscale physics (e.g., satellite altimetry resolution is too coarse) would be welcome background information in the introduction. Is there an advantage to using the numerical model to study how the physics affect real chlorophyll blooms as opposed to high-frequency radar observations in the area? Would SWOT be useful to corroborate the results in the future?

3. Missing definitions:
   - Line 68: Can the authors describe and define how they compute the Chl gradients?
   - Line 72: Can the authors define surface vorticity and be more specific about how they're measuring it? E.g., what depth is "surface", and how are the values calculated from the model? Zeta/f in Fig. 2 is a normalized relative vorticity and should be defined. Equations and more details would be welcome.
   - Line 74: I suggest the authors define "mesoscale" here, e.g. provide a length scale to distinguish from submesoscale.
   - Figures 1 & 5: Can the authors briefly describe how the 95% confidence intervals were computed?
   - Figure A4: Can the authors define horizontal divergence? How do they compute it from the model? An equation would be welcome.

4. The authors present their conclusions in rather qualitative ways, but a few quantitative statistics to support their findings would go far to strengthen the manuscript. Here are some cases where I feel this is relevant:
   - Abstract Line 11: "this submesoscale horizontal stirring mechanism is responsible for ~24% of the seasonal surface chlorophyll increase in the region." The authors mention this result once in the manuscript (Line 69), so it may be easy for the reader to miss. To be presented as a main result in the abstract, I believe it needs more quantitative development. This result is based on a 9-year chl timeseries (2010-2019) average, but the model timeframe is much shorter. This result doesn't account for variability in chl or submesoscale stirring from year to year. I suggest that the authors at least comment on

the range of this value by quantifying the chl increase each year. It could also be mentioned in the discussion that there may be interannual variability in submesoscale stirring. Also, can the authors comment on whether the results are sensitive to increasing the "East" and "West" box sizes?
- Line 94-97: Can the authors provide the statistics for "little differences", "substantially higher", "similar values"?
- Line 101: Can the authors provide the statistics for "substantially wider and more skewed"?
- Figure A5: Can the authors provide the statistics for "good agreement" and "no significant difference"?

5. Figure 1, Panel (a):
   - The circulation patterns are difficult to see in this figure. Can the arrows be plotted at a lower density and larger so that it is easier to see the major directions of flow?
   - It would be helpful for the reader if the boxes in panel (a) were colored in the same colors as panels (b) and (c), and labeled "West" and "East" directly above the boxes
   - I recommend showing the climatologies for spring and winter separately, such that panel (a) becomes two panels. That might help to show how the average circulation patterns change between the two seasons, which would supplement the narrative.
   - Can the authors comment on how the chlorophyll enrichment line was chosen and why they didn't compute the chlorophyll enrichment factor across the central diagonal of the box?

6. Figure 2:
   - The difference in resolution (1000m and 300m) of different parts of the domain is mentioned for the first time in the caption here. Reading sequentially, this is confusing, given that the model setup is not described until much later in the text. This confusion would be alleviated if the Methods Section & Appendix A were moved before the Results.
   - Can the authors adjust the colorscale minimum so that more of the variability is visible in panel (a)? It currently feels like dynamics are hidden because it is all washed out.
   - For panels (b) and (d), can the authors either make the color scales the same, or comment in the caption on why they are different (e.g., "the limits of the colorbars are different in each panel in order to visualize the respective dynamics in the different seasons")

7. Lines 95-99: The authors claim that the modeled vertical velocity, vertical mixing rate coefficients, and vertical gradients in nutrients "all have similar values in the upstream and downstream regions". However, the captions of Figures A2 and A3 provide more context than this. I suggest including Figs A2 and A3 as panels (c) and (d) in Figure 3. Then, the details in the captions could be moved to the main text as well: "Contrary to the observations (Figs. 1- 3 in the main text), this would imply higher Chl concentrations and gradient magnitude should exist west of the Nile Delta if vertical transport processes dominated Chl distribution." and "Contrary to the observations (Figs. 1- 3 in the main text), this would imply higher Chl concentrations and gradient magnitude should exist west of the Nile Delta if vertical mixing processes dominated Chl distribution." These points are important in the logical process of ruling out potential mechanisms other than the

horizontal stirring in driving the winter bloom, so I do not think they should be buried in the Appendix.

8. The Lagrangian particle tracking demonstration is a nice visual to bolster the authors' arguments that submesoscale horizontal stirring from the coast to offshore is increased in the winter. However, I believe that providing more detail would improve the manuscript:
   - What is the sensitivity of the placement of the red box? How was that location chosen?
   - "About 40000 tracer particles were released uniformly" (Lines 209-210): Can the authors provide the exact amount? Does uniformly mean gridded? At what resolution were they seeded?
   - "Advected for 40 days during the winter and 33 days during the summer" (Line 211): Why not advect them for the same amount of time to be able to compare between the seasons? It feels particularly odd in Figure 5 a to have mismatching timescales provided
   - Are the particle distributions sensitive to the date of initialization?
   - Could the authors quantify what fraction of particles enter the "East" box of Figure 1 in each Lagrangian experiment? This would be a nice way to provide a quantitative assessment of Figure 4, and link directly to the region where chlorophyll has been measured. The "East" box could be plotted in Fig. 4 to complement such a calculation.

9. Figure 6 & Lines 142-148: The authors speculate that water particles accumulate on fronts, which move offshore, and then break up. However, the particles in Figure 4d-f appear to be transported by submesoscale instabilities and chaotic stirring, which need not necessarily be persistent fronts. Can the authors clarify their assumptions more and justify the inclusion of Figure 6? If the particles are in fact accumulating on fronts, perhaps plotting the particles with a color gradient in Figure 4 would make that clearer. Or, the authors could measure FTLE and FSLE and measure accumulation along those features. Alternatively, I think it would be justifiable to remove Figure 6 entirely since it is not currently particularly relevant to the key results, and the ideas presented could be left as discussion points.

**Technical corrections**

Lines 1-2: "The large seasonal increase in marine photosynthetic organisms - i.e., phytoplankton bloom - is a ubiquitous oceanic phenomenon…" to "The large seasonal increases in marine photosynthetic organisms - i.e., phytoplankton blooms - are a ubiquitous oceanic phenomenon…"

Line 3: Consider changing "and that supports the growth and development of larger organisms throughout the marine ecosystem" to "and supports the growth of larger marine organisms" to be more concise and avoid a run-on sentence.

Line 4-5: "front and filament circulation patterns…" to "fine-scale frontal and filamental circulations" (where frontal and filamental are adjectives describing the circulations) or to "fine-scale fronts and filaments" (where fronts and filaments are the nouns); I'm also suggesting to add the term "fine-scale" here for defining the submesoscale

Line 5: "characterizing the" to "characteristic of the"

Line 6: "are manifested by" to "manifest as"

Line 7: "are intensified by" to "are also intensified by". Adding "also" here would make it clear that you are suggesting horizontal stirring is another mechanism besides vertical mixing that enhances chl, rather than the only mechanism.

Line 9: Consider using an alternative word to "interior" because that is often taken to mean the deep ocean. Maybe replace "in the sea interior" with "offshore" or "open sea". The word "interior" is used several other times in this way (Lines 74, 83, 87, 99, 110, 138), which I suggest changing as well.

Line 10: "A comparison of" should point to two things. For example, "A comparison of the climatological circulation patterns and chlorophyll time series…" or , "A comparison of spring and winter chlorophyll indicates…"

Line 13: The term "regulating" is unclear. I recommend removing it.

Line 15: "phytoplankton bloom" to "phytoplankton blooms". In my opinion, this is pretty repetitive with the first line of the abstract, and you don't need to define phytoplankton blooms twice. Maybe make more concise like: "Seasonal phytoplankton blooms occur worldwide, playing a critical role in…"

Lines 18-19: This sentence in its current placement jumps the gun a bit. I suggest moving it to the first line of the paragraph at Line 28. Then, the first sentence can be the opener for the paragraph that follows in Line 20.

Line 21: "a proxy to" to "a proxy for"

Lines 26-27: Citation needed for the statement: "in the oligotrophic nutrient-depleted subtropics Chl exhibits a moderate increase driven by enhanced vertical mixing during winter"

Line 40-41: "imagery of Chl concentration that allows monitoring seasonal changes" to "imagery of Chl concentration that allows the monitoring of seasonal changes"

ines 40-42: The text alludes to a "high-resolution" model with the "same spatiotemporal scales" as the satellite data, without providing actual numbers. Can the authors briefly comment on the specific resolution of the satellite and numerical model here? I know there are more details at the end of the text, but it feels odd not to explicitly state the resolution up front.

Line 48: "the increase in nutrient availability resulting from it" to "the resulting increase in nutrient availability"

Lines 59-60: "Analysis of the large-scale spatial variations in Chl distribution reveals that the transition zone between coastal and pelagic waters varies between different parts of the EMS (Fig. 1a)." This sentence is vague and unclear. I'd suggest removing it entirely and starting the paragraph with the sentence "Focusing on the vicinity…"

The authors interchangeably use the terms "northeast", "north-east", and "east" (similarly for west). Consistent terms should be used to avoid confusion. In my opinion, sticking to "east" and "west" would be the clearest. Examples:

- Line 62: "the region to the north and to the east" to "the east region"
- Line 67: "one to the north-west and one to the north-east" to "one to the west and one to the east"; the terms used in this sentence should match the terms used in Figure 1
- Line 69: "north-eastern" to "eastern"
- Line 70: "north-western" to "western"
- Figure 1: "northeast" and "northwest" used in caption; I suggest changing to "east" and "west", as they're referred to in panels (b) and (c)

Line 77: The authors switch from EMS to "Levantine Basin" here. I suggest using EMS throughout, or be sure to explain/define the Levantine Basin.

Lines 85-87: "In contrast, the more uniform distribution of surface Chl during winter suggests that the transport barrier weakens substantially due to an increase in submesoscale activity…" A "uniform distribution of surface Chl" does not inherently suggest that "the transport barrier weakens substantially due to an increase in submesoscale activity". I recommend rewording to something like this: "In contrast, the more uniform distribution of surface Chl during winter suggests that the transport barrier weakens substantially. Here we test the hypothesis that this is due to an increase in submesoscale activity…"

Line 88-89: "It is well documented that submesoscale currents are characterized by strong vertical motions that can amplify nutrient transport and consequently lead to phytoplankton blooms (Mahadevan, 2016; Lévy et al., 2018). To test whether the observed increase in open-sea Chl gradients in the region downstream from the Nile Delta is indeed driven by local effects of the submesoscale dynamics…" The combination of these two sentences leads the reader to believe you're testing if the vertical motions of submesoscale currents increase chl, but you're trying to argue that it's the horizontal stirring. I suggest rewording.

Line 92: I suggest replacing "the two aforementioned open-sea regions" with "the east and west regions highlighted in Figure 1a".

Figure 4: Can the authors add the phrase "horizontal resolution" to avoid confusion with depth? E.g. "Summer (300 m horizontal resolution)"

Line 120: "further quantified", suggest removing "further" because you have not yet quantified the offshore transport at this point, only provide a qualitative analysis

Line 123: Suggest changing "changing" to "constrained"

Line 124: "in 3km winter" to "in the 3km winter"

Line 124: Suggest changing "changes" to "fluctuations"

Line 140: Add a comma between "bloom" and "it"

Figure 5: I suggest labeling the 5b y-axis as "Chlorophyll enrichment factor" instead of "Enrichment Value" to be consistent with the main text. Fig A6 is only referenced in the Fig 5 caption. I suggest referencing it in main text as well.

Line 149: "As previously shown" to "As shown in previous works"; the current wording could be interpreted to mean this result was shown earlier in the text

---

## Author Comment (AC2)

Response letter to the reviewers for: "Submesoscale horizontal stirring enhances seasonal enrichment of low-chlorophyll surface waters in the Eastern Mediterranean."

We thank the reviewers for their constructive comments and suggestions. The authors feel that the suggestions from the reviewers have significantly improved the paper, mainly by pointing out that more quantitative evidence and stronger statistical analyses are needed to back our claims. Additionally, as both reviewers suggested, we restructured the paper so that the Methods section and parts of the Appendix appear before the results. We have addressed all the points raised and implemented the suggested changes, including replacing some of the figures, adding new ones, and modifying the manuscript to improve its clarity and consistency. We hope that the manuscript is now ready for publication in Ocean Science. Our detailed point-by-point response to the reviewers' comments and suggestions is found below. All page/line/reference/figure numbers refer to the clean version of the revised manuscript. Reviewer 1's comments are in blue, our responses are in red. Reviewer 2's comments are in dark red, and our responses are in blue.

Reviewer 1

1. Title: Given the context of the manuscript, I understand the authors' intended meaning for the phrase "seasonal enrichment of low-chlorophyll surface waters". But, I think it could be clearer. The word "enrichment" describing the "low-chlorophyll surface waters" could be interpreted to mean that horizontal stirring makes chlorophyll even lower in the region. "Seasonal" is also vague, considering that the study focuses on winter blooms. Some alternative suggestions are: - "Coastal to offshore submesoscale horizontal stirring enriches wintertime phytoplankton blooms in the oligotrophic Mediterranean Sea" "Submesoscale horizontal stirring increases offshore chlorophyll concentrations during the winter bloom in the Eastern Mediterranean Sea."
Thanks for your suggestion, the authors have changed the title to: Coastal-to-offshore submesoscale horizontal stirring enhances wintertime phytoplankton blooms in the ultra-oligotrophic Eastern Mediterranean Sea.
2. The authors tackle the challenge of explaining satellite chlorophyll observations with the dynamics observed in a numerical model. To increase trust that they're not comparing "apples and oranges", I suggest that they include more information about the validation of the numerical model and how representative it may be of real-world physics observed in the system. A few sentences would be sufficient, potentially citing more of the findings of the original study by Solodoch et al. 2023 (by the way, I think this study is accidentally cited as two separate citations in this manuscript). Additionally, an explanation for why a numerical model, as opposed to observations, was needed to capture the submesoscale physics (e.g., satellite altimetry resolution is too coarse) would be welcome background information in the introduction.

Is there an advantage to using the numerical model to study how the physics affect real chlorophyll blooms as opposed to high-frequency radar observations in the area? Would SWOT be useful to corroborate the results in the future?

We thank the reviewer for the suggestion. In the revised manuscript, we have included more details about the model and have discussed the validation of the model briefly (L75-99).

Justification for the use of this specific satellite product and the importance of the model usage in this research has been added to the introduction (L44-55). Additionally, a passage regarding the possible future implementations of SWOT, its advantages and disadvantages has been added to the Discussion section (L283-287).

3. Missing definitions: - Line 68: Can the authors describe and define how they compute the Chl gradients?

An explanation has been added in the Methods section (subsection PDFs) (L124-126), including an equation.

Line 72: Can the authors define surface vorticity and be more specific about how they're measuring it? E.g., what depth is "surface", and how are the values calculated from the model? Zeta/f in Fig. 2 is a normalized relative vorticity and should be defined. Equations and more details would be welcome.

Thank you for the suggestion. In the revised manuscript, we have clarified that 'surface' corresponds to the 2-m depth and have also defined relative vorticity and horizontal divergence (L110-114).

Line 74: I suggest the authors define "mesoscale" here, e.g. provide a length scale to distinguish from submesoscale.

Thank you, a definition has been added (L175).

Figures 1 & 5: Can the authors briefly describe how the 95% confidence intervals were computed?

Thank you, a description has been added to the Method section, subsection Chlorophyll Climatologies (L120).

Figure A4: Can the authors define horizontal divergence? How do they compute it from the model? An equation would be welcome.

Thank you for the suggestion. Horizontal divergence has been defined at L110, including an equation.

4. The authors present their conclusions in rather qualitative ways, but a few quantitative statistics to support their findings would go far to strengthen the manuscript. Here are some cases where I feel this is relevant: - Abstract Line 11: "this submesoscale horizontal stirring mechanism is responsible for ~24% of the seasonal surface chlorophyll increase in the region." The authors mention this result once in the manuscript (Line 69), so it may be easy for the reader to miss. To be presented as a main result in the abstract, I believe it needs more quantitative development. This result is based on a 9-year chl timeseries (2010-2019) average, but the model timeframe is much shorter. This result doesn't account for variability in chl or submesoscale stirring from year to year. I suggest that the authors at least comment on the range of this value

by quantifying the chl increase each year. It could also be mentioned in the discussion that there may be interannual variability in submesoscale stirring. Also, can the authors comment on whether the results are sensitive to increasing the "East" and "West" box sizes?

Thank you for this comment. Statistics for the observed enhancement range and interannual variability of the Chl concentration have been added to the results section (L159-170) as well as to the discussion (L254-258). Also, the authors have further clarified what the 24% enhancement is in relation to.

Additionally, a few sentences have been added about the box size sensitivity in the Methods section (we tested the calculation with a 50% box size increase) (L165-170).

Line 94-97: Can the authors provide the statistics for "little differences", "substantially higher", and "similar values"?

Thank you for this comment. The authors have added the Kolmogorov–Smirnov test results for the comparison between East and West for summer and winter in the results section and removed these phrases from the text (L196-199).

Additionally, the statistical comparison between the nutriclines has been added to the Figure caption (A3) in the appendix.

Line 101: Can the authors provide the statistics for "substantially wider and more skewed"?

Thank you for this suggestion. The authors have removed this phrase and added the requested statistics to support this claim (L195-209).

Figure A5: Can the authors provide the statistics for "good agreement" and "no significant difference"?

Thank you for this suggestion. This phrase has also been removed and the statistics added (L196) along with further interpretation of the PDFs (L201-209).

5. Figure 1, Panel (a): - The circulation patterns are difficult to see in this figure. Can the arrows be plotted at a lower density and larger so that it is easier to see the major directions of flow?  - - - It would be helpful for the reader if the boxes in panel (a) were colored in the same colors as panels (b) and (c), and labeled "West" and "East" directly above the boxes I recommend showing the climatologies for spring and winter separately, such that panel (a) becomes two panels. That might help to show how the average circulation patterns change between the two seasons, which would supplement the narrative. Can the authors comment on how the chlorophyll enrichment line was chosen and why they didn't compute the chlorophyll enrichment factor across the central diagonal of the box?

Thank you for this comment. The authors have split Figure 1A into two figures for the mean summer distribution of Chl and then the mean winter distribution. The current vectors have been plotted less densely and in a brighter color. The boxes for east and west have been plotted in the same colors as panel C and with titles above them, as suggested.

Additionally, the authors have removed previous Figure 1C (Chl gradient climatology) as it wasn't really adding much, wasn't mentioned in the text and better portrayed through the PDFs.

Regarding the enrichment factor, the diagonal we chose was based on the following considerations:

The starting point is relatively in the center of the Chl rich region and also allows for a relatively linear increase of distance from the coast. If we commence the diagonal further to the north, then as we increase distance from the Israeli coast, the distance to the Egyptian north coast decreases and vice versa.

The endpoint is the area in which the regional minimum chl values were measured per season.

The relationship between the diagonal and the East box isn't necessarily important; the horizontal transport enriches the whole eastern domain. The authors have added this explanation to the Methods section, subsection CEF (L138-141).

6. Figure 2:  - The difference in resolution (1000m and 300m) of different parts of the domain is mentioned for the first time in the caption here. Reading sequentially, this is confusing, given that the model setup is not described until much later in the text. This confusion would be alleviated if the Methods Section & Appendix A were moved before the Results.

- - Can the authors adjust the colorscale minimum so that more of the variability is visible in panel (a)? It currently feels like dynamics are hidden because it is all washed out. For panels (b) and (d), can the authors either make the color scales the same, or comment in the caption on why they are different (e.g., "the limits of the colorbars are different in each panel in order to visualize the respective dynamics in the different seasons").

Thank you for these suggestions, the authors have moved the Methods Section and combined it with portions of the Appendix so that it appears before the results section and have also added, as suggested in one of the minor comments, both the chl grid and model resolutions in the introduction ("We combine multi-year satellite imagery of Chl concentration (1km gridded) that allows the monitoring of seasonal changes in phytoplankton biomass down to submesoscales with a nested high-resolution (3km, 1km and 300m) circulation model that resolves complex ….") (L54-56).

Regarding the color scales, the authors have changed the panels (A) and (C) to a logarithmic color scale so that it matches Figure 1. We cannot have the same color scales for A,B,C, and D since (B) and (D) are diverging scales (so zero is centered) while (A) and (C) are sequential.

7. Lines 95-99: The authors claim that the modeled vertical velocity, vertical mixing rate coefficients, and vertical gradients in nutrients "all have similar values in the upstream and downstream regions". However, the captions of Figures A2 and A3 provide more context than this. I suggest including Figs A2 and A3 as panels (c) and (d) in Figure 3.

Then, the details in the captions could be moved to the main text as well: "Contrary to the observations (Figs. 1- 3 in the main text), this would imply higher Chl concentrations and gradient magnitude should exist west of the Nile Delta if vertical transport processes dominated Chl distribution." and "Contrary to the observations (Figs. 1- 3 in the main text), this would imply higher Chl concentrations and gradient magnitude should exist west of the Nile Delta if vertical mixing processes dominated Chl distribution." These points are important in the logical process of ruling out potential mechanisms other than the horizontal stirring in driving the winter bloom, so I do not think they should be buried in the Appendix.

Thank you for these suggestions. The figures A2 and A3 have been combined as suggested in Figure 3, and the explanations previously found in the captions in the appendix have been moved to the Methods section (L132) and to the Results (L199-209). Additionally, several lines outlining the logical process have been added (L199-200, 209, 213, 216-217) to further clarify our rationale.

8. The Lagrangian particle tracking demonstration is a nice visual to bolster the authors' arguments that submesoscale horizontal stirring from the coast to offshore is increased in the winter. However, I believe that providing more detail would improve the manuscript:  - What is the sensitivity of the placement of the red box? How was that location chosen? - - - - "About 40000 tracer particles were released uniformly" (Lines 209-210): Can the authors provide the exact amount? Does uniformly mean gridded? At what resolution were they seeded? "Advected for 40 days during the winter and 33 days during the summer" (Line 211): Why not advect them for the same amount of time to be able to compare between the seasons? It feels particularly odd in Figure 5 a to have mismatching timescales provided.

Are the particle distributions sensitive to the date of initialization?  Could the authors quantify what fraction of particles enter the "East" box of Figure 1 in each Lagrangian experiment? This would be a nice way to provide a quantitative assessment of Figure 4, and link directly to the region where chlorophyll has been measured. The "East" box could be plotted in Fig. 4 to complement such a calculation.

Thank you for your comments. In our revised manuscript, we have made it clear that exactly 40,000 particles are released in a 200 x 200 grid pattern (L106). As explained in the manuscript,  the particle advection during summer is selected to coincide with the period of boundary current instabilities and spiral formations that begins in the later half of the month of July. Subsequently, we only had 33 days of velocity field from the model solution to advect the particles during summer.

The particle experiments have been designed to highlight the differences in the physical processes involved in the offshore transport tracers flowing through the boundary current during summer and winter seasons.

These experiments track the tracer particles released at a given time over a small patch in the Nile Delta region, the majority of which get advected by the boundary current. The particle experiments do not fully capture the continuous advection of the nutrient-rich Nile Delta water through the boundary current. Consequently, the sampling of tracer particles arriving at the east and the west boxes will be highly time dependent. Nevertheless, as demonstrated in Figures 4a-I and 5a, the offshore transport of the tracer is considerably different. In both seasons, the released particles get swept by the boundary current. However, in summer, their lateral transport is dictated mainly by the advection through the boundary current and the spiral system, whereas in winter, the horizontal stirring by the submesoscale current also plays a significant role. Moreover, these experiments allow us to quantify the extent of the offshore transport of the tracer flowing through the boundary current for about a month in both seasons, as shown in Figure 5a.

9. Figure 6 & Lines 142-148: The authors speculate that water particles accumulate on fronts, which move offshore, and then break up. However, the particles in Figure 4d-f appear to be transported by submesoscale instabilities and chaotic stirring, which need not necessarily be persistent fronts. Can the authors clarify their assumptions more and justify the inclusion of Figure 6? If the particles are in fact accumulating on fronts, perhaps plotting the particles with a color gradient in Figure 4 would make that clearer. Or, the authors could measure FTLE and FSLE and measure accumulation along those features. Alternatively, I think it would be justifiable to remove Figure 6 entirely since it is not currently particularly relevant to the key results, and the ideas presented could be left as discussion points.

Thank you for this suggestion. As suggested by additional reviewers, Figure 6 has been removed since we haven't included enough analyses to justify its inclusion.

Technical corrections:
Lines 1-2: "The large seasonal increase in marine photosynthetic organisms - i.e., phytoplankton bloom - is a ubiquitous oceanic phenomenon…" to "The large seasonal increases in marine photosynthetic organisms - i.e., phytoplankton blooms - are a ubiquitous oceanic phenomenon…"
This has been amended in the text.
Line 3: Consider changing "and that supports the growth and development of larger organisms throughout the marine ecosystem" to "and supports the growth of larger marine organisms" to be more concise and avoid a run-on sentence.
This has been amended in the text.
Line 4-5: "front and filament circulation patterns…" to "fine-scale frontal and filamental circulations" (where frontal and filamental are adjectives describing the circulations) or to "finescale fronts and filaments" (where fronts and filaments are the nouns); I'm also suggesting to add the term "fine-scale" here for defining the submesoscale
This has been amended in the text (L4-5). A sub-mesoscale definition has been modified at L27.

Line 5: "characterizing the" to "characteristic of the"

This has been amended in the text.

Line 6: "are manifested by" to "manifest as"

This has been amended in the text.

Line 7: "are intensified by" to "are also intensified by". Adding "also" here would make it clear that you are suggesting horizontal stirring is another mechanism besides vertical mixing that enhances chl, rather than the only mechanism.

This has been amended in the text.

Line 9: Consider using an alternative word to "interior" because that is often taken to mean the deep ocean. Maybe replace "in the sea interior" with "offshore" or "open sea". The word "interior" is used several other times in this way (Lines 74, 83, 87, 99, 110, 138), which I suggest changing as well.

Thank you, I've limited the term to either open sea or pelagic.

Line 10: "A comparison of" should point to two things. For example, "A comparison of the climatological circulation patterns and chlorophyll time series…" or "A comparison of spring and winter chlorophyll indicates…"

Thank you, this has been amended in the text. We changed it to "Comparing the 2010–2019 chlorophyll time series from two pelagic regions equidistant from the Nile Delta…" (L58).

Line 13: The term "regulating" is unclear. I recommend removing it.

Thank you, the authors have changed to "modulating".

Line 15: "phytoplankton bloom" to "phytoplankton blooms".

In my opinion, this is pretty repetitive with the first line of the abstract, and you don't need to define phytoplankton blooms twice. Maybe make more concise like: "Seasonal phytoplankton blooms occur worldwide, playing a critical role in…"

This has been amended in the text.

Lines 18-19: This sentence in its current placement jumps the gun a bit. I suggest moving it to the first line of the paragraph at Line 28. Then, the first sentence can be the opener for the paragraph that follows in Line 20.

This has been amended in the text.

Line 21: "a proxy to" to "a proxy for"

This has been amended in the text.

Lines 26-27: Citation needed for the statement: "in the oligotrophic nutrient-depleted subtropics Chl exhibits a moderate increase driven by enhanced vertical mixing during winter"

Thank you for this suggestion, references to Siokou-Frangou et al., 2010; Barale et al., 2008 have been added to the text.

Line 40-41: "imagery of Chl concentration that allows monitoring seasonal changes" to "imagery of Chl concentration that allows the monitoring of seasonal changes"

This has been amended in the text.

Lines 40-42: The text alludes to a "high-resolution" model with the "same spatiotemporal scales" as the satellite data, without providing actual numbers. Can the authors briefly comment on the specific resolution of the satellite and numerical model here? I know there are more details at the end of the text, but it feels odd not to explicitly state the resolution up front.

Thank you, yes, we've changed the line to:"we combine multi-year satellite imagery of Chl concentration (1km gridded) that allows the monitoring of seasonal changes in phytoplankton biomass down to submesoscales with a nested high-resolution (3km, 1km and 300m) circulation model that resolves complex …." (L53)

Line 48: "the increase in nutrient availability resulting from it" to "the resulting increase in nutrient availability"

This has been amended in the text.

Lines 59-60: "Analysis of the large-scale spatial variations in Chl distribution reveals that the transition zone between coastal and pelagic waters varies between different parts of the EMS (Fig. 1a)." This sentence is vague and unclear. I'd suggest removing it entirely and starting the paragraph with the sentence "Focusing on the vicinity…" The authors interchangeably use the terms "northeast", "north-east", and "east" (similarly for west). Consistent terms should be used to avoid confusion. In my opinion, sticking to "east" and "west" would be the clearest.

Thank you for this suggestion. This line has been removed.

Examples: Line 62: "the region to the north and to the east" to "the east region" - - - - - Line 67: "one to the north-west and one to the north-east" to "one to the west and one to the east"; the terms used in this sentence should match the terms used in Figure 1 Line 69: "north-eastern" to "eastern" Line 70: "north-western" to "western" Figure 1: "northeast" and "northwest" used in caption; I suggest changing to "east" and "west", as they're referred to in panels (b) and (c)

Thank you, we've narrowed it down to East and West as suggested or occasionally upstream-downstream when the distinction was important.

Line 77: The authors switch from EMS to "Levantine Basin" here. I suggest using EMS throughout, or be sure to explain/define the Levantine Basin.

Thank you for this suggestion; all mentions of the Levantine have been changed to EMS.

Lines 85-87: "In contrast, the more uniform distribution of surface Chl during winter suggests that the transport barrier weakens substantially due to an increase in submesoscale activity..." A "uniform distribution of surface Chl" does not inherently suggest that "the transport barrier weakens substantially due to an increase in submesoscale activity". I recommend rewording to something like this: "In contrast, the more uniform distribution of surface Chl during winter suggests that the transport barrier weakens substantially. Here we test the hypothesis that this is due to an increase in submesoscale activity..."

This has been amended in the text.

Line 88-89: "It is well documented that submesoscale currents are characterized by strong vertical motions that can amplify nutrient transport and consequently lead to phytoplankton blooms (Mahadevan, 2016; Lévy et al., 2018). To test whether the observed increase in open-sea Chl gradients in the region downstream from the Nile Delta is indeed driven by local effects of the submesoscale dynamics…" The combination of these two sentences leads the reader to believe you're testing if the vertical motions of submesoscale currents increase chl, but you're trying to argue that it's the horizontal stirring. I suggest rewording.

Thank you for this suggestion, the authors have rephrased this passage to make the statement clearer (L191-195).

Line 92: I suggest replacing "the two aforementioned open-sea regions" with "the east and west regions highlighted in Figure 1a".

This has been amended in the text.

Figure 4: Can the authors add the phrase "horizontal resolution" to avoid confusion with depth? E.g. "Summer (300 m horizontal resolution)"

This has been amended in the text.

Line 120: "further quantified", suggest removing "further" because you have not yet quantified the offshore transport at this point, only provide a qualitative analysis

This has been amended in the text.

Line 123: Suggest changing "changing" to "constrained"

This has been amended in the text.

Line 124: "in 3km winter" to "in the 3km winter"

This has been amended in the text.

Line 124: Suggest changing "changes" to "fluctuations"

This has been amended in the text.

Line 140: Add a comma between "bloom" and "it"

This has been amended in the text.

Figure 5: I suggest labeling the 5b y-axis as "Chlorophyll enrichment factor" instead of "Enrichment Value" to be consistent with the main text.

This has been amended in the text.

Fig A6 is only referenced in the Fig 5 caption. I suggest referencing it in the main text as well.

Thank you, this has been amended in the text and Methods (L115).

Line 149: "As previously shown" to "As shown in previous works"; the current wording could be interpreted to mean this result was shown earlier in the text.

This has been amended in the text.

Reviewer 2

The authors present the role of lateral exchanges by sub-mesoscale instabilities in controlling the productivity of the interior of the Eastern Mediterranean Sea. In contrast to previous studies, which have typically focused on the vertical velocities, the authors focus on horizontal transports between the productive coastal and Nile outflow influenced regions and the oligotrophic interior. Overall, I found the paper convincing and worthy of publication in Ocean Science; however, I have a few suggestions below to improve the paper.

I found the structure of the manuscript inconsistent with Ocean Science style papers and made it (at times) not very smooth to read. I regularly found myself jumping from parts of section 2 to section 4 and appendix A in order to understand the model and data sets used. I would suggest restructuring so that the methods and content in appendix A come before the results. I would also suggest strengthening the methods with more discussion of the model, the inputs (river and topography seems like they would be very important for this but are not described in the paper) and its validation to provide confidence in the results.

Thank you for this suggestion. We moved the Methods section so that it appears before the Results and combined it with most of the content previously presented in the Appendix. Following another reviewer's suggestion, we combined several PDFs that were previously included in the Appendix into a single four-panel figure (Figure 3). Additional information regarding the model validation and setup has been added to the Methods section (L75-99). For more in-depth descriptions regarding the model and its validation, please see Solodoch et al. 2023.

I was a little confused around line 96 –97. If I understood this line correctly the model has a bio-geo-chemical component (e.g. the authors mention modelled nutrient gradients) but we are not shown any BGC results from the model. It would seem very important to see if the model used in the later analysis reproduces the satellite derived seasonal variability.

Thank you for pointing this out. The main model (CROCO) used in the paper does not have a BGC component.

In order to verify that there are no innate nutrient gradient differences in the mixed layers in the east and west regions, we used the Copernicus global ocean biogeochemical hindcast product, which uses the PISCES biogeochemical model. See figure A3 in the appendix section and BGC model data in the methods section. Just in case I've made it more clear in the text as well (L213)

Fig 6 and the associated text is not really demonstrated in this work. If the point is to be made it should be tested in the paper. Would it be possible using the particle tracking experiment. Can you demonstrate that the particles are accumulating on fronts prior to leaving the boundary region? Then becoming more evenly distributed once further off shore?

Thank you for this suggestion. Figure 6 has been removed entirely (as suggested by another reviewer).

Minor Comments

Line 10 to 11 > I don't think that the statement about 24% of the winter enhancement being driven by sub-mesoscale motions is supported by the manuscript. The only reference I could find is that the eastern box has 24% higher chl than the western box in winter. It seems a bit of a jump to then state that this is (1) all driven by submesoscale flows and (2) imply it applies to the whole region.

Thank you for this comment. We've added more statistics in order to clarify and strengthen this argument, including further analysis of the PDFs (L195-210). Additionally, we've added several lines to outline the logic and help clarify our thought process for the reader (L199-200, 209, 213, 216-217).

Line 145 > Missing the "2)".

This has been amended in the text.

---

## Referee Report (RR1)

Minor Comments:

1. Lines 91-95: It is unclear how the authors are drawing these conclusions here. Are these findings all from Solodoch et al. 2023? The wording is vague, e.g., "good agreement", "appear consistently", "generally in agreement", "compare well", "fair agreement". I recommend briefly clarifying how these conclusions were drawn with more quantitative language.

2. Lines 162-165: "When referenced to the shared summer minimum, the wintertime increase in the East is ~24% larger than in the West…" I believe that it is important to make this finding crystal clear since it is included in the abstract as the main quantitative conclusion. Is the "shared summer minimum" the minimum mean Chl concentration of the two boxes, or is it the minimum from the unsmoothed daily Chl data? Is the 24% increase the mean result? Are 11.5 (Line 164) and 25% (Line 165) mean values? Where are the results of 24% compared to 25% drawn from? 25% and not 24% is referenced again in the Discussion (Line 258).

3. The authors clearly show that the submesoscale circulations increase offshore transport by comparing Lagrangian particle simulations in the 300m and 3km wintertime simulations. However, the 3km offshore particle transport is still larger than the summertime (Figures 4B/H, C/I, and 5A), indicating that mesoscale transport plays some role and should not be entirely discounted. Lines 233-235 would be an appropriate place to further clarify this, e.g. "***mesoscale* wintertime spirals**", with some mention in Lines 259-261. E.g., "the observed winter enhancement cannot be explained by local vertical processes, regional differences in nutricline structure***, or mesoscale currents alone***."

Technical Corrections:

1. Abstract Line 6: Recommend removing the word "also", since that is interpreted to reference the previous sentence, which describes vertical submesoscale processes that are not the diagnosed mechanism of increasing chlorophyll in this case.

2. Line 18: Recommending not to start a new paragraph here to improve flow.

3. Line 81: Readers may not be familiar with what a sigma level is. Please briefly define, e.g. "terrain-following depth levels".

4. Lines 138-139: Reference the dashed lines in Figure 1A, B to visually aid the reader, and describe in the caption of Figure 1 what the dashed line represents.

---

## Author Response (AR2)

Response letter to the reviewers for: "Coastal-to-offshore submesoscale horizontal stirring enhances wintertime phytoplankton blooms in the ultra-oligotrophic Eastern Mediterranean Sea"

We thank the reviewer for the careful reading of the manuscript and for the constructive comments. In response, we have clarified the methodology, refined the presentation of the results, and added quantitative explanations where appropriate. All changes are detailed below. The reviewers' comments are in black, and the authors' responses are in dark red.

Minor Comments:

1. Lines 91-95: It is unclear how the authors are drawing these conclusions here. Are these findings all from Solodoch et al. 2023? The wording is vague, e.g., "good agreement", "appear consistently", "generally in agreement", "compare well", "fair agreement". I recommend briefly clarifying how these conclusions were drawn with more quantitative language.

We thank the reviewer for this comment. The validation of the circulation model is presented in detail in Solodoch et al. (2023), which is cited here as the primary reference for model realism and performance. We agree that the original wording in this manuscript was overly qualitative. To address this, we have revised the text to explicitly attribute the validation results to Solodoch et al. (2023), along with referencing specific figures from that paper and the validation of a variety of parameters. Additionally, we replaced qualitative expressions with specific, quantitative statements reported in that study where possible.

In particular, we now state that the domain-mean surface eddy kinetic energy differs from altimetry-derived estimates by approximately 0.002 m2 s−2 and from CMEMS reanalysis by approximately 0.005 m2 s−2,  and note that spatial smoothing (to fall in line with available observation resolutions) improves agreement between modeled and observed EKE fields. We also clarify that modeled sea-surface temperature and salinity differences relative to satellite and reanalysis products fall within the statistical uncertainty of those datasets, and that the domain-mean mixed layer depth was almost identical to the reanalysis. Finally, we specify that the 300-m model frequency spectra are in fair agreement with observations from the DeepLev mooring and that it captures the observed wintertime submesoscale energization in the 1–5 day band above 500 m depth. Monthly mean velocities are in agreement with a shelf break-mooring (Rosentraub and Brenner 2007) (L 89-100).

We believe these revisions clarify the basis for our statements while appropriately referring the reader to Solodoch et al. (2023) for the full validation analysis.

2. Lines 162-165: "When referenced to the shared summer minimum, the wintertime increase in the East is ~24% larger than in the West…" I believe that it is important to make this finding crystal clear since it is included in the abstract as the main quantitative conclusion. Is the "shared summer minimum" the minimum mean Chl concentration of the two boxes, or is it the minimum from the unsmoothed daily Chl data? Is the 24% increase the mean result? Are 11.5 (Line 164) and 25% (Line 165) mean values? Where are the results of 24% compared to 25% drawn from? 25% and not 24% is referenced again in the Discussion (Line 258).

The authors thank you for this observation. The shared summer minimum has been better defined as "To compare the seasonal amplitude between regions, winter Chl concentrations were referenced to the common summer baseline defined by the mean summer Chl concentration. " (L 167), and we added another line for clarity regarding the averaging:" For each region, Chl values represent spatial averages over the respective boxes and temporal averages over the indicated seasons" (L 163).

The mismatched reports stem from a rounding inconsistency. The exact figures calculated from the 10-year time series are:
Winter mean concentrations (Jan–Mar):
  East: 0.06665 mg/m³
  West: 0.05974 mg/m³
  East is 11.56% higher than West (reported as 11.6%)

Summer mean concentrations (Jul–Sep):
  East: 0.02792 mg/m³
  West: 0.02794 mg/m³
  East is 0.09% lower than West (reported as 0.1%)

Winter increase in the East referenced to the summer baseline, is 24.76% (reported as 24.8%)

Interannual Variability:
Winter EN>WN %  : mean = 11.529%,  std = 8.886% (reported as 11.5% and 8.9%)
Extra EN rise % : mean = 24.953%,  std = 22.471% (reported as 25.0% and 22.5%)

The ten-year mean winter difference between the east and west (11.56%) is different from the mean derived from the interannual variability (11.529%) because its calculated differently. The former is a percentage computed from **10-year mean values,** and the latter is the **mean of yearly percentages.** It's the same for the seasonal amplitude calculations and statistics. At some point, we must have rounded 24.7% down to 24% instead of 24.8%. We have now ensured that all statistics are reported consistently and that the correct numbers have been added to the abstract and the results section.

3. The authors clearly show that the submesoscale circulations increase offshore transport by comparing Lagrangian particle simulations in the 300m and 3km wintertime simulations. However, the 3km offshore particle transport is still larger than the summertime (Figures 4B/H, C/I, and 5A), indicating that mesoscale transport plays some role and should not be entirely discounted. Lines 233-235 would be an appropriate place to further clarify this, e.g. "mesoscale wintertime spirals", with some mention in Lines 259-261. E.g., "the observed winter enhancement cannot be explained by local vertical processes, regional differences in nutricline structure, or mesoscale currents alone."

The authors thank you for this comment. As suggested, we have added clarification that indeed mesoscale structures play a significant part in offshore transport. (L 239,267).

Technical Corrections:

1. Abstract Line 6: Recommend removing the word "also", since that is interpreted to reference the previous sentence, which describes vertical submesoscale processes that are not the diagnosed mechanism of increasing chlorophyll in this case.
Thank you, the word "also" has been removed.
2. Line 18: Recommending not to start a new paragraph here to improve flow.
Thank you. The authors have implemented your suggestion for line 18.
3. Line 81: Readers may not be familiar with what a sigma level is. Please briefly define, e.g. "terrain-following depth levels".
Thank you for pointing this out. The sentence has been rephrased and a definition added: "...employing 80, 120, and 150 terrain-following (σ) vertical levels…" (L80).
4. Lines 138-139: Reference the dashed lines in Figure 1A, B to visually aid the reader, and describe in the caption of Figure 1 what the dashed line represents
Thank you. The authors have added a reference to Figure 1 at Line 142, and added a description of what the line represents in the caption for Figure 1.